# Immunization against Gonadotropin-Releasing Hormone in Female Beef Calves to Avoid Pregnancy at Time of Slaughter

**DOI:** 10.3390/ani11072071

**Published:** 2021-07-12

**Authors:** Julia Schütz, Jürn Rudolph, Adrian Steiner, Esther Rothenanger, Jürg Hüsler, Gaby Hirsbrunner

**Affiliations:** 1Nutztierpraxis Rudolph AG, CH-6280 Hochdorf, Switzerland; JuliaSchuetz@gmx.de (J.S.); j.rudolph@seetalvet.ch (J.R.); 2Clinic for Ruminants, Vetsuisse Faculty, University of Bern, Bremgartenstrasse 109 a, CH-3012 Bern, Switzerland; adrian.steiner@vetsuisse.unibe.ch; 3labor-zentral.ch, Stationsweg 3, CH-6232 Geuensee, Switzerland; esther.rothenanger@labor-zentral.ch; 4Institute of Mathematical Statistics and Actuarial Science, University of Bern, Sidlerstrasse 5, CH-3012 Bern, Switzerland; juerg.huesler@stat.unibe.ch

**Keywords:** cattle, anti-gonadotropin-releasing hormone (anti-GnRH), Improvac^®^, immunocastration, puberty, progesterone, cortisol

## Abstract

**Simple Summary:**

Precociousness of heifers kept in mixed beef herds with young and adult bulls leads to unwanted pregnancies. Inbreeding, premature calving followed by dystocia and a high stillbirth rate are the consequences. As an alternative, such heifers are slaughtered during the resulting pregnancy. The slaughtering of pregnant animals is an ethical problem, which is strongly criticized by consumers and animal welfare organizations. Therefore, the aim of this study was to postpone puberty in female beef calves housed in Swiss mixed herds to avoid pregnancy until scheduled slaughter at 11 months of age. We used a vaccine (Improvac^®^) that induces antibody production against sexual hormones, thereby suppressing the reproductive cycle. Monthly progesterone analysis in blood was performed to decide whether the cycle had already started. The results proved, that vaccinated female beef were not coming in heat until 11 months of age compared to the unvaccinated control group, which came in heat earlier. In conclusion, vaccination with Improvac^®^ is an animal-friendly, non-invasive and reliable method to avoid early pregnancy in heifers as well as the slaughter of pregnant cattle.

**Abstract:**

Precocious puberty in beef heifers can result in unwanted pregnancies due to accidental breeding by farm bulls. Inbreeding, premature calving followed by dystocia and a high stillbirth rate or slaughtering of pregnant heifers are the consequences of this behaviour. The aim of the study was to postpone puberty by using Improvac^®^, an anti-GnRH vaccine. Therefore, *n* = 25 calves were twice vaccinated, once at the age of 5 and then at 6.5 months. *n* = 24 calves served as unvaccinated case controls. The onset of puberty was assigned if progesterone analysis in the blood exceeded 1 ng/mL. Progesterone values were excluded if the corresponding serum cortisol levels were ≥60 nmol/L. Our target was met, as in the vaccinated group none of the calves exceeded a progesterone value >1 ng/mL until the scheduled age of slaughter at 11 months and only 12.5% of the animals exceeded a progesterone value of 1 ng/mL over the whole measuring period (>400 days) compared with 56.5% of the calves in the control group. In conclusion, the favourable results from our study using the vaccine Improvac^®^ represent an animal-friendly, non-invasive and reliable way to avoid early pregnancy in heifers as well as the slaughter of pregnant cattle.

## 1. Introduction

The herd size of beef cow-calf operations in Switzerland are small where male and female animals of all age groups are housed together. There is often no possibility to split up the herd (only one free stall housing available, not many different pastures per farm). This often leads to young heifers being inadvertently impregnated by farm bulls. Animal welfare and ethics are of concern when these young pregnant heifers are presented at slaughter [1]. A German cross-sectional study revealed that slaughtering pregnant cattle is a widespread practice with many fetuses in the second or third trimesters [1]. Due to increased consumer concern, a study in a Swiss abattoir was initiated that reported a pregnancy prevalence of 5.67% cattle pregnant > 5 months and 0.67% 7 to 9 months pregnant (BLV: Projekt Schlachtung von trächtigen Rindern-Prävalenz und Gründe der Schlachtung). Based on these results from that study, at present, farmers have to declare the pregnancy status in cows later than 5 months post-partum (p.p.) and heifers older than 15 months when slaughtered (Proviande: Fachempfehlung zur Vermeidung des Schlachtens von trächtigen Tieren der Rindviehgattung).

Puberty in a calf is defined as a measure of a physical development leading to sexual maturity. In cattle, puberty mostly occurs between 8 and 12 months of age, but there is a large variation depending on the breed [2,3]. Therefore, declaring pregnancy status at an age > 15 months of age might be too late in early mature beef heifers. The onset of puberty in calves is defined as the age at first ovulation and an increase in plasma progesterone concentrations above 1 ng/mL [4]. As rectal palpation is not feasible in calves/young heifers, analysis of serum progesterone might be used to determine the beginning of puberty (serum levels above 1 ng/mL [4,5]). However, it was also demonstrated in ovariectomised cattle, that under a certain stress condition, animals might secrete additional progesterone (1.8 ± 0.7 ng/mL) and cortisol (11.5–44.3 ng/mL (≈31.7–122.2 nmol/L)) from the adrenal cortex [6].

In dairy heifers, the early onset of puberty is one of the breeding goals to reduce costs for the rearing period [7]. Different studies focused on intensifying the feeding of calves in the rearing period in order to advance the age at puberty [3,8,9,10]. In beef heifers intended for slaughter, though, precocious puberty is not desired due to the risk of unwanted pregnancy. In addition, in replacement heifers, inadequate skeletal maturity can be a problem if the age at first calving is <24 months [11] resulting in increased rates of dystocia [3].

If puberty in calves should irreversibly be avoided, surgical methods, such as ovariectomy might be used [12,13]. As an alternative, the suppression of the hypothalamo-gonadal axis using an anti-GnRH vaccine is an animal-friendly option that is even reversible [14,15,16]. Immunocastration suppresses sexual behaviour in male and female cattle, sheep, pigs and horses [17,18,19,20,21,22,23,24]. It also reduces aggressive or undesirable behavior of pigs [25] mares [26] and bulls [27].

Our goal was to postpone puberty in beef calves by using an anti-GnRH vaccine at 5 and 6.5 months of age compared to untreated calves from the same herd. Consequently, the vaccinated group should not be pregnant when slaughtered at ±11 months.

## 2. Materials and Methods

### 2.1. Animals, Care and Housing

*n* = 49 female calves from 14 farms were included in this study (*n* = 24 in the control group (C) and *n* = 25 in the vaccinated group (V)). The breeds were Limousin, Angus, Swiss Fleckvieh and mixed breeds. All herds consisted of female and male calves with breeding bulls and teaser bulls and were housed in pens with straw bedding or cubicle housing systems. The cattle had free access to water and to pasture, depending on weather and temperature. Diets consisted of grass, hay, corn and/or grass silage. Data were collected from May 2019 to August 2020.

### 2.2. Treatment with Anti-GnRH Vaccine

Improvac^®^ is an anti-GnRH vaccine (Zoetis Schweiz GmbH, 2800 Delémont, Switzerland) containing an analog of GnRH linked to a carrier protein combined with a synthetic aqueous adjuvant (200 μg of GnRH-protein-conjugate per mL). Heifer calves in group V, received 2 doses (initial and booster) of Improvac^®^ 6 weeks apart (initial vaccination was at 5 months ± 14 days). The dosage used for both the initial and the booster vaccination was 400 μg of GnRH-protein-conjugate (2 mL of Improvac^®^). All injections were administered subcutaneously on the right side of the neck. In group C, the animals received 2 mL of 0.9% saline solution subcutaneously on the right side of the neck (twice, 6 weeks apart).

All animal experimentation was performed with permission and in accordance with Swiss law. The following approval number was allocated by the animal experimentation commission (elected by the cantonal executive council): BE 73/19.

### 2.3. Including and Excluding Criteria

Calves were randomly assigned to either group (1–2 calves per group and farm). Before the injections, the calves were subjected to basic clinical examination including rectal body temperature, heart rate, respiratory rate, auscultation of lungs and gastrointestinal tract, and presence of umbilical remnants. If the rectal temperature was >39.5 °C and/or auscultation revealed pathological findings (bronchopneumonia, audible heart anomalies), the respective calf was excluded. After vaccination, the injection site was inspected for signs of tissue reaction by the farmers twice daily. If swelling was observed and if appetite decreased (food intake subjectively judged by farmers) to ≤75% of the normal amount, the veterinarian was notified.

### 2.4. Blood Sampling, Progesterone and Cortisol Values

In all calves, blood was sampled during both vaccination visits and then every 4 weeks until slaughter or until the end of the study (4–9 samples/animal). The blood samples were collected by venipuncture from the jugular vein or the V. caudalis mediana into serum tubes (S-Monovetten 9 mL with Clot Activator, Sarstedt, Nümbrecht, Deutschland). After a clotting time of 1–3 h at room temperature, samples were centrifuged (4000× *g*, 10 min), and serum was then stored at −18 °C for later analysis. Progesterone concentration was measured by chemiluminescence assay on an Immulite 2000 XPi (Siemens Healthcare, Zürich, Switzerland). The analytical sensitivity was 0.1 ng/mL and the measurement range was 0.2–40 ng/mL. The assay was performed according to the manufacturer’s instructions. Cortisol concentration was measured by chemiluminescence assay on an Immulite 2000 XPi (Siemens Healthcare, Zürich, Switzerland). The analytical sensitivity was 5.5 nmol/L and the measurement range was 0.99–1380 nmol/L. The assay was performed according to the manufacturer’s instructions. If serum cortisol levels were >60 nmol/L [28] the corresponding serum progesterone value was excluded from further analysis.

### 2.5. Statistical Analysis

The primary endpoint was the number of calves not revealing a cycle at the time of slaughter (10–12 months of age). The study was planned as a case-control study with an assumption of 20% of the calves vaccinated confirmed in the cycle and 80% of the not vaccinated calves in the cycle (difference 60%) and a power to be at least 80%. The number of calves per group was calculated to be *n* = 25, each (resulting power = 88%). For categorical data, the frequency of categories was determined. For metric variables, mean, median, sd, 25% and 75% quantiles, minimum and maximum were calculated. Survival times were estimated by Kaplan–Meier estimation and compared by a frailty model because the data were clustered within a farm. The events of a progesterone value above 1 ng/mL during the whole measuring period were compared between the two groups with a generalised linear mixed model. The McNemar test was applied for the frequency table whether a vaccinated calf and a control calf within a farm had a progesterone value above 1 ng/mL during the whole measuring period.

A *p*-value < 0.05 indicated a significant result. Data were analysed using the statistical software SAS^®^ version 9.4 (SAS Institute Inc., Cary, NC, USA).

## 3. Results

### 3.1. Treatment with Anti-GnRH Vaccine

Age at first vaccination/placebo administration was 157 d in group V (median; 25%/75% quartiles: 151 d/164 d) compared to 159 d (148 d/169 d) in group C. Median heart rate, respiratory rate and rectal body temperature at first and second vaccination and duration of the period between vaccinations are listed on Table 1. Age at slaughter was comparable in both groups (median 326 days in group V compared to 320 days in group C).

### 3.2. Side Effects, Including and Excluding Criteria

Slight swelling at the injection site as is described to occur frequently after vaccination with Improvac^®^ [14] was not observed. Reduction of food intake ≤75% of the normal amount judged subjectively by farmers was observed in *n* = 1 calf of group V for 1 day (without elevated rectal body temperature). Retrospectively, a pair of calves (*n* = 1 from group V, *n* = 1 from group C, same farm) had to be excluded because all measured serum cortisol values exceeded 60 nmol/L [28]. In group V, a total of 27/169 progesterone samples in serum had to be excluded (corresponding cortisol sample > 60 nmol/L). In group C, a total of 15/164 progesterone samples in serum had to be excluded (corresponding cortisol sample > 60 nmol/L [28]). The farmers in this study subjectively judged herds to be calmer, even if only one of a pair of calves was vaccinated. They also mentioned that fewer accidents were occurring.

### 3.3. Progesterone and Cortisol Values

Differences in progesterone concentrations during the study period are shown in Figure 1a,b and Table 2. The number of calves with progesterone concentration above 1 mg/mL varied between V and C groups during the study period (*p* = 0.0028). In group V only 3/24 animals (12.5%) exceeded a progesterone value of 1 ng/mL in all samples measured (age: 346 d, 362 d, 363 d), whereas there were 13/23 (56.5%) calves in group C with a progesterone value of >1 ng/mL, beginning at the age of 286 d. The Logrank test (*p* = 0.0038) showed a highly significant difference between calves within groups V and C, versus the duration of the period between second vaccination and progesterone value > 1 ng/mL (Figure 2).

Median (25th/75th quartiles) cortisol values with included progesterone values in group V and C were 18.4; 8.4/35.3 nmol/L and 21.5; 8/36.9 nmol/L, respectively. Cortisol values with excluded progesterone values (>60 nmol/L) in group V and C were 97.7 (81/115.8) nmol/L and 86.8 (65.8/108.8) nmol/L, respectively.

### 3.4. Age at Slaughter

The target age at the slaughter of the calves of this study was 11 months. In group V *n* = 16 calves were slaughtered at 326 days (median), but there were *n* = 7 animals slaughtered at a median age of 617 days (1 animal still alive). In group C, *n* = 17 calves were slaughtered at 320.5 days (median), but there were *n* = 4 animals slaughtered at a median age of 603 days (3 animals still alive). The reason for keeping these animals longer was due to better meat prices (special meat label).

## 4. Discussion

Our objective was achieved since none of the calves in group V exceeded a progesterone value > 1 ng/mL until the age of 11 months (scheduled age of slaughter). Furthermore, only 12.5% of the animals in group V exceeded a progesterone value of 1 ng/mL during the entire period of measurement compared with 56.5% of the calves in group C. Different studies already proved cycle suppression in adult cows [14,15,29,30] and reduction of serum levels of testosterone in bulls resulting in decreased sexual and aggressive behaviour [20,31,32,33]. The most simple and reliable parameter for clinical veterinarians in the field to predict a return to estrus after vaccination was the discovery of class III follicles (>9 mm) [14,34]. However, in growing calves, rectal palpation is not feasible. Different authors relied on progesterone values to define the beginning of puberty [4,35]. Progesterone values, however, are not reliable if concomitant cortisol values in serum are high (>60 nmol/L [28]), as cortisol and progesterone then originate from the adrenal glands. This was demonstrated in dehorned steers [36], white-tailed deer [37] and also in a cold stress test in women [38]. We therefore included only progesterone values if cortisol values were ≤60 nmol/L [28]. Weekly or biweekly progesterone analyses would have been preferable. The decision for blood sampling every 4 weeks was a concession to logistics and the farmers’ time expenditure and the risk of handling the animals living in herds. It cannot be excluded that a progesterone value > 1 ng/mL was possibly missed.

Vaccination with an anti-GnRH vaccine of very young animals (2, 6 and 13 weeks of age) could not postpone the age of puberty in cattle (52–54 weeks) compared to unvaccinated control animals [30]. Alternatively, some studies were performed in older heifers with a bodyweight of 230–480 kg [29,39,40], which was too late for the objective of our study. We decided to vaccinate calves at 5 and 6.5 months of age intending to postpone their puberty until the age of 11 months. If heifers are kept longer on farms, a third vaccination should be performed to extend the time until the beginning of puberty (as performed in cattle to extend the period of cycle suppression [16]), but this was not in the present study’s perspective. The influence of the presence of bulls on age at puberty was another discussion point, as there were adult bulls in all farms included. Neither a study in young heifers (140–402 d of age) [41] nor in heifers of 287 d of age [42] indicated that the presence of an adult bull altered the incidence of precocious puberty.

The vaccine Improvac^®^ was used, as in Europe the cattle-specific vaccine Bopriva^®^ is not allowed due to the preservative Thiomersal. Therefore, the use of Improvac^®^ (preservative Chlorocresol) was allowed for cattle by the Institute of Virology and Immunology (responsible for the approval and monitoring of animal vaccines and immune sera for use in veterinary medicine in Switzerland).

In Switzerland, the number of beef cows is increasing compared to the decreasing number of dairy cows (Agristat–Statistik der Schweizer Landwirtschaft 2021). Therefore, an increasing number of pregnant young heifers at slaughter is expected. Additionally, interactions among animals in heat can lead to accidents and impair carcass quality [43]. It was described that cattle during estrus were mounted four times in stables and seven times in pasture [44]. Farmers stated that with an increasing number of heifers in heat, herds were more restless, suffered more injuries and were more difficult to handle. In conclusion, the favourable results from our study represent a step forward in solving this problem in an animal-friendly, non-invasive and reliable way to avoid early pregnancy in heifers as well as the slaughter of pregnant cattle.

## Figures and Tables

**Figure 1 animals-11-02071-f001:**
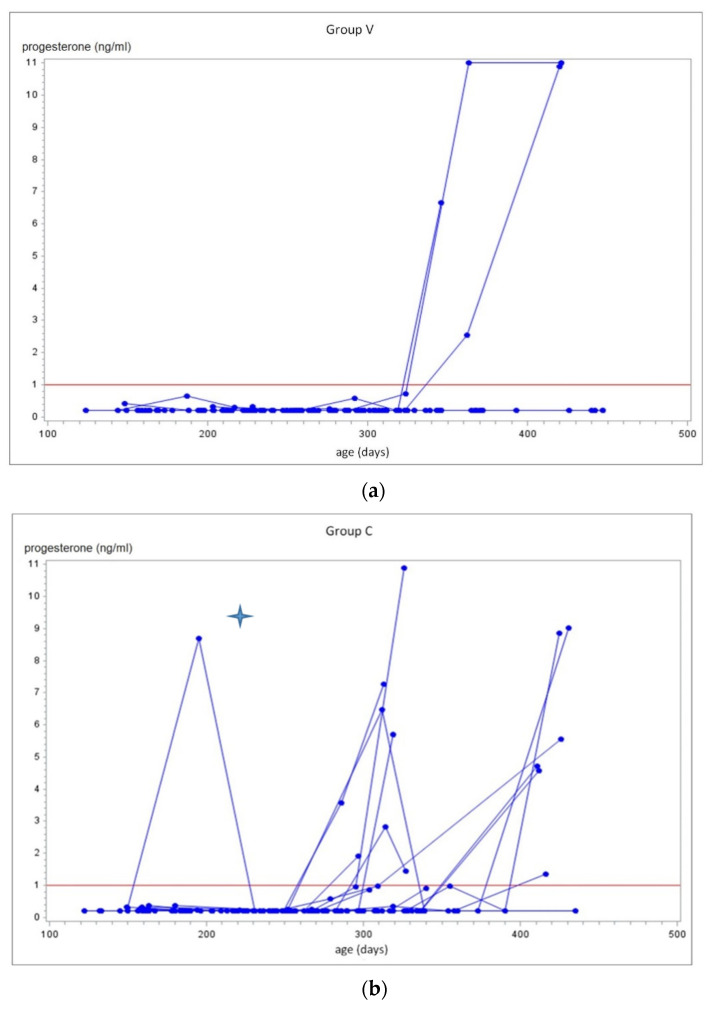
(**a**) Age of the calves in group V and their progesterone values over time. The red line represents the progesterone value of 1 ng/mL in serum. Only three calves exceeded a progesterone value of 1 ng/mL. (**b**) Age of the calves in group C and their progesterone values over time. The red line represents the progesterone value of 1 ng/mL in blood. The youngest heifer exceeding the progesterone value of 1 ng/mL was 286 d old. This animal had a progesterone value of 8.7 ng/mL at the second injection. The corresponding cortisol value was 57.1 nmol/L and close to our threshold of 60 nmol/L.

**Figure 2 animals-11-02071-f002:**
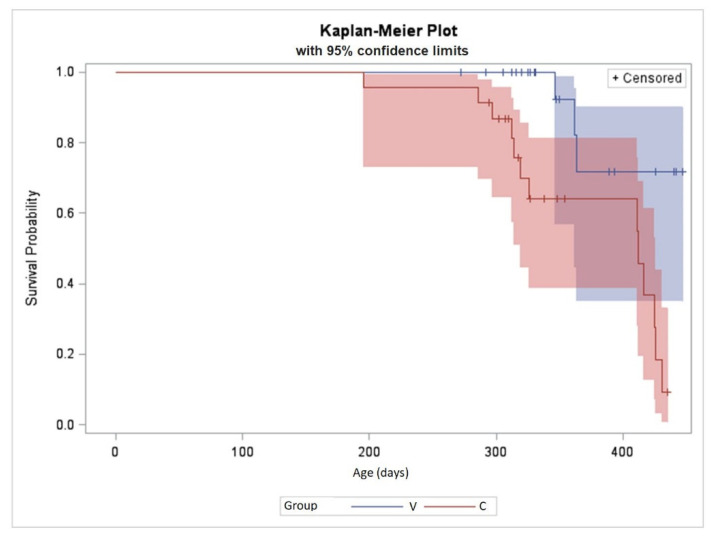
Kaplan–Meyer graph representing calves within groups V (blue) and C (red), versus the duration of the period between second vaccination and progesterone value > 1 ng/mL (Logrank test *p* = 0.0038) including the confidence limits of these data. + means censored data (calves slaughtered).

**Table 1 animals-11-02071-t001:** Heart rate in beats/min; respiratory rate in breaths/min; rectal body temperature in °C. Age at slaughter of *n* = 32 slaughtered calves available. Median values are indicated with 25%/75% quartiles in brackets.

Parameters	Improvac^®^ (V)	Control (C)
First vaccination		
Heart rate	114 (85/128)	128 (104/140)
Respiratory rate	32 (28/36)	40 (36/44)
Rectal body temperature	39.0 (38.8/39.4)	39.2 (38.8/39.5)
Second vaccination		
Heart rate	110 (91.5/132)	120 (84/136)
Respiratory rate	32 (28/40)	30 (28/39)
Rectal body temperature	39.0 (38.8/39.3)	38.8 (38.6/39.0)
Period between vaccination (days)	42 (39/46.5)	41 (39/47)
Age at slaughter (days)	326 (312/349.3)	320.5 (307.8/336.8)

**Table 2 animals-11-02071-t002:** The McNemar test for the frequency table whether a vaccinated calf and a control calve within a farm had a progesterone value above 1 ng/mL during the whole measuring period revealed a significant difference (*p* = 0.0047).

Progesterone (C)	Progesterone (V)		
	**0**	**1**	**Total**
**0**	3	0	3
**1**	8	2	10
**Total**	11	2	13

## Data Availability

Excel tables including all data are available at the corresponding author. Statistical analyses are available at the corresponding author.

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
