# Peer review of "Immunization against Gonadotropin-Releasing Hormone in Female Beef Calves to Avoid Pregnancy at Time of Slaughter"

_animals, 2021, doi:10.3390/ani11072071_

Round 1
Reviewer 1 Report
I have read the paper with pleasure. The experiment is well designed, well explained and the conclusions have a great interest for the practitioner and the farmer. Congratulations for the authors.
I have just a question at the line 44 : could you give some reasons explaining the difficulties to split the herd ?
Author Response
Reviewer 1: Thank you very much for your interest in this subject and for reviewing our manuscript.
Line 44: we added 2 reasons "only 1 free stall housing available, not many different pastures per farm"
Reviewer 2 Report
The evaluated study deals with the hormonal contraception used to postpone puberty by immunization against GnRH in young female beef calves. This issue is, to some extent, interesting, because presents a new way how to avoid the unwanted pregnancies in young female heifers kept together with male ones. It should be stressed, that this breeding system is a specific condition in small mix farms occurring in a few countries. Due to this, the outcomes of this study might be interesting mainly for readers in these countries.
However, the outcomes of this experiment are worth being published, because this study develops something, that has been rather weakly explored until now.
The general study design, with exception of sampling protocol for progesterone, is rather satisfactory. Manuscript is well-structured and clearly written. Due to this, this paper is easy to follow and understand. In general, the conclusions are supported by the obtained data and the Authors used an adequate clinical and laboratory methods.
However, I have some concerns regarding the sampling protocol for progesterone measurements. The samples were obtained only once monthly; that means too rare for this experiment, because normal bovine estrus cycle lasts about 3 weeks. In addition, it is known that at puberty ovarian cycling is very often irregular and includes a numerous short cycles. Due to this, it is possible that in this study not all progesterone rises reflecting ovarian activity were detected. In my opinion, the Authors should consider this aspect and implement such a remark into Discussion. But to be honest, I have to add that the endpoint of this study was to minimize the number of unwanted pregnancies in young heifers and this result is correct. Moreover, to my knowledge, Improvac is not registered for the application in the bovine. Because the Authors recommend the use of this vaccine in the practice, this information should be also given in the paper.
In conclusion, I recommend this paper for publication after the two above mentioned revisions.
Author Response
Thank you very much for your interest in our manuscript and for its review.
Of course, you are right as to the frequency of progesterone analyses. It was just very difficult to convince farmers to frequent blood sampling - and the animals were not easy to handle at all. The following sentences were included into the discussion as explanation: "Weekly or biweekly progesterone analyses would have been preferable. The decision for blood sampling every 4 weeks was a concession to logistics and the farmers' time expenditure and the risk of handling the animals living in herds. It cannot be excluded that one or other progesterone value > 1ng/ml was missed."
Furthermore, a passus on Improvac was included: "
The vaccine Improvac® was used, as in Europe the cattle-specific vaccine Bopriva® is not allowed due to the preservative Thiomersal. Therefore, the use of Improvac® (preservative Chlorocresol) was allowed for cattle by the Institute of Virology and Immunology (responsible for the approval and monitoring of animal vaccines and immune sera for use in veterinary medicine in Switzerland)."
Reviewer 3 Report
Manuscript ID: animals-1292678
Title: Immunization against gonadotropin-releasing hormone in female beef calves to avoid pregnancy at time of slaughter
Authors: Julia Schütz , Jürn Rudolph , Adrian Steiner , Esther Rothenanger , Jürg Hüsler , Gaby Hirsbrunner
The aim of the study was to postpone puberty by using Improvac® , an anti-GnRH vaccine. He study was performed on 49, 25 vaccinated and 24, unvaccinated calves The heifer calves were vaccinated twice at 5 and 6.5 months of age. Serum progesterone concentrations were used as measure of cyclicity.
The sample size, study design and methods were clear and meet the requirements for addressing the objective.
However, sampling frequency is questionable. The sampling must be done every week or at least every 2 weeks to determine cyclicity. In the study blood was collected during both vaccination visits and then every 4 weeks until slaughter or until the end of the study. Please provide reference or justification why 4 weeks was determined.
Different timing of vaccination will add additional information to the literature.
Language correction is needed.
Due to delaying of puberty and unwanted pregnancy the following need to be clarified.
- Whether all farms had proven bulls to offer equal risk for impregnation?
- Biostimulation (presence of bulls) and its effect on age at puberty should be considered. Inclusion of this in discussion will add strength to the paper favoring the objective.
Incidence of precocious puberty in beef heifers should be included.
Ref:
Wehrman ME, Kojima FN, Sanchez T, Mariscal DV, Kinder JE. Incidence of precocious puberty in developing beef heifers. J Anim Sci. 1996 Oct;74(10):2462-7. doi: 10.2527/1996.74102462x.
Line 23: Change “und” to “and”
Line 25 – 26: Change to “Precocious puberty in beef heifers can result in unwanted pregnancies due to accidental breeding by farm bulls”.
Lin 36-37: State the conclusion of study findings. Please avoid opinions
Line 41 to 44: Sentences could be changed to….Herd size of beef cow calf operation in Switzerland is small where male and female animals of all age groups are housed together. This often lead to young heifers being inadvertently impregnated by farm bulls. Animal welfare and ethics are of concern when these young pregnant heifers presented at slaughter.
Line 48 to 54: Due to increased consumer concern, a study in Swiss abattoir was initiated that reported a pregnancy prevalence of 5.67% in > 5 months age group and 0.67% for 7 to 9 months age group. Based on the results from that study, at present, Swiss abattoir demands pregnancy status for heifers >15 mo of age and for cows > 5 mo post calving.
Line 55: “measure” instead of “part”
Line 54-55: remove or move to first line of next paragraph.
Line 62: remove “define” and insert “determine”
Line 64: remove “challenge’ and insert “condition”
Line 66: Be specific. “In dairy heifers,….”
Line 67-68: Rephrase the sentences.
“Different studies focused on intensifying feeding of calves in the rearing period in order to advance the age at puberty [3, 9-10]”. In beef heifers intended for slaughter though, precocious puberty is not desired due to the risk of unwanted pregnancy. In addition, in replacement heifers, inadequate skeletal maturity can be a problem if the
age at first calving is < 24 months [11] resulting in increased rates of dystocia [3].
If puberty in calves should irreversibly be avoided, surgical methods, such as ovariectomy might be used [12-13]
Line 83: Limousin, Angus, Swiss Fleckvieh and mixed breeds.
Due to age at puberty differ between breeds mixed breeds should be clarified.
Line 84-85: All herds consisted female and male calves, with breeding bulls and teaser bulls and were housed in pens with straw bedding or cubicle housing systems.
Line 87: Diets consisted of grass, hay, corn and / or grass silage.
Line 92-94:
Heifer calves in group V, received 2 doses (initial and booster) of Improvac® 6 wks apart (initial vaccination was at 5 months ± 14 days).
Line 103-106: Before the injections, the calves were subjected to basic clinical examination including rectal body temperature, heart rate, respiratory rate, auscultation of lungs and gastrointestinal tract, and presence of umbilical remnants.
Study period was 5 to 11 months. Blood sample was collected every 4 weeks. Line 114 says 4 to 9 samples /animal. Clarify how often blood was collected for the calves had 4 samples during the study period.
Line 208: Behavior. This was not clarified in materials and methods or in the analysis except injection site reaction and fever. Was a questionnaire/survey presented to the producers?
Figure 2 was not cited in the text.
X axis, Days since vaccination would be addressing to the objective instead of age (days) of calves. Please revise.
Rephrase the sentence 225-227; It seems that 56.5% in C group had > 1 ng for the entire period. But they had > 1 ng before 11 mo (286 days). This is important and need to be specified.
Line 230: Presence of CL is more reliable than class III follicles. Rephrase.
Line 227-228: cycle suppression in bulls? Please clarify.
Line 234: This was demonstrated….
Line 245: but this was not in the present study’s perspective.
Line 250: remove “compromise” and insert “impair”
Conclusion is missing.
Line 280: Evans, A.C.; Rawlings, N.C.,….remove “,”
Formatting of references need to be in accordance with journal recommendation.
Author Response
Dear reviewer,
many thanks for your constructive comments and English improvements. You find all our comments below
Sampling of blood: Of course, you are right as to the frequency of progesterone analyses. It was just very difficult to convince farmers to frequent blood sampling - and the animals were not easy to handle at all. The following sentences were included into the discussion as explanation: "Weekly or biweekly progesterone analyses would have been preferable. The decision for blood sampling every 4 weeks was a concession to logistics and the farmers' time expenditure and the risk of handling the animals living in herds. It cannot be excluded that one or other progesterone value > 1ng/ml was missed."
Delaying of puberty and unwanted pregnancy:
- All farms had adult bulls in the herds, only 1 farm with teaser bulls (artificial insemination)
- The following passus was included in the discussion: " The influence of the presence of bulls on age at puberty was another discussion point, as there were adult bulls in all farms included. Neither a study in young heifers (140-402d of age) [41], nor in heifers of 287d of age [42] indicated, that the presence of an adult bull al-tered the incidence of precocious puberty." And the following references added:
Wehrman, M. E.; Kojima, F. N.; Sanchez, T.; Mariscal, D. V.; Kinder, J. E. Incidence of precocious puberty in developing beef heifers. J Anim Sci 1996, 74, 2462-67.
Roberson, M. S.; Ansotegui, R. P.; Berardinelli, J. G.; Whitman, R. W.; McInerney, M. J. Influence of biostimulation by mature bulls on occurrence of puberty in beef heifers. J Anim Sci 1987, 64, 1601-05.
Figure 2: A sentence and a citation in the text was introduced: " The Logrank test (p=0.0038) showed a highly significant difference between calves within groups V and C, versus the duration of the period between second vaccination and pro-gesterone value >1ng/ml (Figure 2)." As the goal of our work was to have the animals slaughtered at 11 months of age without cycling (and getting pregnant), we would like to keep the age on the x-axis.
Line 23: done
Lines 25-26: sentence changed according to reviewers' proposal
Lines 36-37: sentence has been changed to: " In conclusion, the favourable results from our study using the vaccine Improvac® represent an animal friendly, non-invasive and reliable way to avoid early pregnancy in heifers as well as the slaughter of pregnant cattle.."
Lines 41-44: paragraph changed according to proposals of Rev 1 and 3: " Herd size of beef cow calf operation in Switzerland is small where male and female animals of all age groups are housed together. There is often no possibility to split up the herd (only 1 free stall housing available, not many different pastures per farm). This often leads to young heifers inadvertently impregnated by farm bulls. Animal welfare and ethics are of concern when these young pregnant heifers presented at slaughter [1]."
Lines 48-54: mostly changed, but it is not the age of > 5 months, and 7-9 months, respectively, but the duration of pregnancy. "Due to increased consumer concern, a study in a Swiss abattoir was initiated that re-ported a pregnancy prevalence of 5.67% cattle pregnant > 5 months and 0.67% 7 to 9 months pregnant (BLV: Projekt Schlachtung von trächtigen Rindern – Prävalenz und Gründe der Schlachtung). Based on these results from that study, at present, farmers have to declare the pregnancy status in cows later than 5 months post partum (p.p.) and heifers older than 15 months when slaughtered (Proviande: Fachempfehlung zur Vermeidung des Schlachtens von trächtigen Tieren der Rindviehgattung)."
Line 55: done
Lines 54-55: removed to next paragraph
Line 62: done
Line 64: done
Line 66: done
Lines 67 and further: changed according to proposal " Different studies focused on intensifying feeding of calves in the rearing period in order to advance the age at puberty [3, 8-10]. In beef heifers intended for slaughter, though, precocious puberty is not desired due to the risk of unwanted pregnancy. In addition, in re-placement heifers, inadequate skeletal maturity can be a problem if the age at first calving is < 24 months [11] resulting in increased rates of dystocia [3]."
Lines 73-74: done
Line 83: Unfortunately mixed breeds are difficult to clarify. Often they come from inseminations with SILIAN which is a mix of semen from Angus, Limousin and Simmental and the mother a HF, Brown Swiss, Original Brown, Luing or Fleckvieh cow. Sometimes already mother cows are mixed. (List of calves' breed-mixes available at author)
Lines 84-85: changed according to proposal
Line 87: done
Lines 92-94: done according to proposal " Heifer calves in group V, received 2 doses (initial and booster) of Improvac® 6 wks apart (initial vaccination was at 5 months ± 14 days)."
Lines 103-106: changed according to proposal: "Before the injections, the calves were subjected to basic clinical examination including rectal body temperature, heart rate, respiratory rate, auscultation of lungs and gastrointestinal tract, and presence of umbilical remnants."
Study period 5-11 months; blood values were mostly taken 6 times (4-9); in both groups there was 1 animal slaughtered already at 272 days / 294 days, respectively (resulting in 4 samples each).
Line 208: Behavior: No, there was no questionnaire presented, but the heifers were from farms supervised by the first author and farmers often expressed their subjective opinion. Has been added to the paragraph "side effects".
Lines 225-227: Thanks for that comment! Hope it is clearer now. "In group V only 3/24 animals (12.5%) exceeded a progesterone value of 1ng/ml in all sam-ples measured (age: 346d, 362d, 363d), whereas there were 13/23 (56.5%) calves in group C with a progesterone value of 1ng/ml, beginning at the age of 286d."
Lines 227-228: changed to: "Different studies already proved cycle suppression in adult cows [14-15, 29-30] and reduction of serum levels of testosterone in bulls resulting in decreased sexual and aggressive behaviour [20, 31-33]."
Line 230: This is true for heifers and start of cycling. In cattle, however, CL's may stay on ovaries once cows are vaccinated. It was one of the conclusions of our study (Balet et al., J Dai Sci 2013) that class III follicles are the best predictive parameter for recurring cycles.
Line 234: done
Line 245: changed according to reviewers' proposal
Line 250: done
References were formatted according to journal recommendation.
Round 2
Reviewer 3 Report
The authors addressed the queries sufficiently.
Line 182-184: Change "Progesterone values over time are visible in Figure 1a,b." to Differences in progesterone concentrations during the study period are shown in Figure 1a and b. Number of calves with progesterone concentration above 1 mg / mL varied between V and C groups during the study period.
Line 210 - 212: Median (25th, 75th quartiles) cortisol values with included progesterone values in group V and C were 18.4 (8.4, 35.3) nmol/l and 21.5 (8, 36.9 nmol/l), respectively. Cortisol values with excluded progesterone values (> 60 nmol/l ) in group V and C were 97.7 (81, 115.8) nmol/l and C: 86.8 (65.8, 108.8) nmol/l, respectively.
Still language can be improved but it is authors preference
Author Response
Dear reviewer,
both your comments have been included in the manuscript. Thank you very much for all your work!
Kind regards
Gaby Hirsbrunner